# Effect of Curcumin Addition on the Properties of Biodegradable Pectin/Chitosan Films

**DOI:** 10.3390/molecules26082152

**Published:** 2021-04-08

**Authors:** Qingtong Xie, Xudong Zheng, Liuting Li, Liqun Ma, Qihui Zhao, Shiyuan Chang, Lijun You

**Affiliations:** 1School of Food Science and Engineering, South China University of Technology, Guangzhou 510640, China; 201864391246@mail.scut.edu.cn (Q.X.); 201892393257@mail.scut.edu.cn (X.Z.); 201865390316@mail.scut.edu.cn (L.L.); 201862390494@mail.scut.edu.cn (L.M.); 201866391367@mail.scut.edu.cn (Q.Z.); 2Overseas Expertise Introduction Center for Food Nutrition and Human Health (111 Center), South China University of Technology, Guangzhou 510640, China

**Keywords:** chitosan, pectin, curcumin, film, DES

## Abstract

A pectin/chitosan matrix-loaded curcumin film (PCCF) with a deep eutectic solvent (DES) as the solvent and plasticizer was prepared in this study. Different quantities of curcumin (identified as PCCF-0, PCCF-1, PCCF-2. PCCF-3) were loaded on the pectin/chitosan film in order to evaluate their effects on the film properties. Results showed that curcumin could interact with the pectin/chitosan matrix and form a complex three-dimensional network structure. PCCF could promote the thickness, tensile strength, thermal properties, antioxidant and antiseptic capacities, but deteriorate the light transmission and elongation at the same time. The addition of curcumin would change the color of the film, without significantly affecting the moisture content. The tensile strength of PCCF-3 reached the maximum value of 3.75 MPa, while the elongation decreased to 10%. Meanwhile, the water-resistance properties of PCCF-3 were significantly promoted by 8.6% compared with that of PCCF-0. Furthermore, PCCF showed remarkable sustained antioxidant activities in a dose-dependent manner. PCCF-3 could inhibit DPPH and ABTS free radicals by 58.66% and 29.07%, respectively. It also showed antiseptic capacity on fresh pork during storage. Therefore, curcumin addition could improve the barrier, mechanical, antioxidant and antiseptic properties of the polysaccharide-based film and PCCF has the potential to be used as a new kind of food packaging material in the food industry.

## 1. Introduction

According to a 2020 FAO report, roughly 750 million people or 10% of the world’s population face food safety problems [1]. Fifty-four percent of the world’s food spoilage occurs during the post-harvest processing and storage period, and 46% takes place during transport and consumption [2]. As a result, food preservation is a huge challenge in the whole world. In order to extend the shelf-life of food, a variety of physical, chemical and biological techniques, including low-temperature storage, controlled atmosphere storage, heat treatment, irradiation treatment and the use of preservatives have been used [3,4,5,6]. In the last decade, alternative food preservative techniques have emerged, considered as promising alternatives to gradually replace the traditional preservation process, especially films with a certain function in food packaging, which has become the research hotspot. Bioactive film packaging exhibits the advantages of retaining gas, holding moisture, antibacterial or/and antioxidant activities and biodegradability properties.

Most studies in food packaging have focused on natural raw materials with excellent film-forming properties, such as proteins, lipids and polysaccharides. The films formed by proteins and lipids have poor transparency and low gloss, which affect the sensory appeal of the products [7,8], however, polysaccharide-based films have excellent gas resistance and gloss but show strong water permeability, which restricts their application in food packaging [9]. In most cases, the addition of multi-component hybrid film would be superior to that prepared by a single component, especially in the mechanical properties, water/oil resistance, heat sealing, antioxidant and antibacterial activities. For example, the tensile strength of a starch film added with cellulose was increased by 18 times, and the water vapor transmission ratio was reduced compared with the starch film without cellulose. Riaz et al. [3] found that blending pectin and chitosan at a ratio of 1:1 increased tensile strength, reduced water solubility, water vapour permeability and oxygen permeability. Therefore, pectin and chitosan composites could be used as a film matrix to improve the tensile strength and barrier properties.

Chitosan is a kind of cationic polysaccharide obtained by deacetylation of chitin extracted from crustacean shells [10], and it has been studied in the field of food active packaging for its sustained drug release [11], biodegradability, bacteriostatic and non-toxic properties [12]. However, the disadvantages such as poor mechanical properties, poor hydrophilic and narrow bacteriostatic spectrum limit its applications. In addition, chitosan only exhibits good solubility under specific acidic conditions, which makes the process of dissolution more complicated. Therefore, it is necessary to overcome its hydrophilic and mechanical property weaknesses. A deep eutectic solvent (DES) is a liquid formed by the interaction of two or more components through strong hydrogen bonds, in which the melting temperature is much lower than that of any of the single components [12,13]. DES is not only cheap, but also has high solubility, low toxicity, low volatility, and good bio-degradability, which has been widely used in bioactive extraction [14], reaction media, melt processing [15], and organic synthesis [16]. In this study, DES was used to dissolve chitosan instead of acid reagents (used in some traditional methods) and had the additional advantage of eliminating the bad odor. Meanwhile, it played an important role in improving the mechanical properties of the film.

Pectin is a typical linear heteropolysaccharide consisting of a large number of negatively charged carboxylic acid groups [17], therefore, it shows good solubility in water and has the possibility of interacting with cations. Previous studies showed that pectin can form a three-dimensional network structure by adding calcium ions, which can fill in the gaps of the twisted pectin chain and improve the strength of the film [18]. It has been reported that chitosan (a cationic polysaccharide) with good film-forming property, has the possibility of cross-linking with pectin (anionic hydrophilic polysaccharide) by electrostatic attraction [18]. Therefore, in this study, pectin and chitosan were used to prepare a polysaccharide film [19]. However, the resulting film has generally weaker bacteriostatic and barrier properties which limit its application based on some previous studies [18]. A common approach to improve film properties is to add other bioactive polymers [20]. Curcumin is a low molecular weight polyphenolic compound isolated from turmeric. It has many pharmacological activities, such as anti-inflammatory, antioxidation, lipid regulation, antibacterial, antifungal and antiviral activities and anti-tumor, with low toxicity and small adverse reactions [21,22,23,24]. Curcumin is mainly used in the coloring of sausage products, canned products, stewed product sauces and other products in the food industry [21,25]. Curcumin has two hydroxyl groups at both ends, which can bind to amino and/or carboxyl groups by hydrogen bonds [18]. Curcumin has been blended with polymers such as cellulose, cellulose acetate, collagen and gelatin when preparing functional films [26,27,28,29], but not many efforts have been reported on the use of curcumin when preparing pectin/chitosan films.

Therefore, the purpose of this study was to evaluate the effects of curcumin addition to pectin/chitosan films. Firstly, varying quantities of curcumin were loaded on pectin/chitosan films. Then, SEM, FT-IR, DSC and XRD analysis were used to characterize the films. Thirdly, the thickness, light transmission, surface color, mechanical properties and water solubility were also studied. Finally, the in vitro antioxidant and antiseptic activities in a food system were evaluated. The results will pave the way for exploring a new food package with antioxidant and antiseptic activities in the food industry.

## 2. Results and Discussion

### 2.1. Characterization of PCCF

The physical property of polysaccharide film is very important to its application. In this study, the thickness, water content, water solubility, surface color, opacity and mechanical properties (including tensile strength and elongation at break) were determined.

#### 2.1.1. Thickness

The average thickness, film color and optical properties of PCCF were presented in Table 1. Film thickness is very important to its mechanical strength and light transmittance [30]. The thickness of the film increased markedly with the addition of curcumin (*p* < 0.05). The maximum thickness is 0.426 ± 0.025 mm, which is 2.29 folds of that of the control.

#### 2.1.2. Film Color and Optical Characterization

The L*, a* and b* values meant lightness, green or red, blue or yellow, respectively. According to Table 1, the film without curcumin was transparent, with the values of L* 49.64, a* 4.31, and b* 30.365. With the addition of curcumin, L* significantly increased to 57.82. In contrast, a* markedly decreased to 0.56 and b* increased to 44.86, implying that the color of the film gradually became yellow.

The surface color and light transmittance of the film played an important role in its application since they could affect the stability of the packaged food and consumers’ senses. The transparency of the film was related to the absorption of light, the compatibility of materials and the thickness. The interface between different materials affected the reflection or scattering of light, further affecting the transparency. Results showed that the transmittance of the film decreased significantly as the curcumin content increased (*p* < 0.05) (Table 1). The transmittance of the pectin/chitosan film (PCCF-0) was 30.27 ± 2.87%, which was nearly four times of that of PCCF-3, indicating that the addition of curcumin had an adverse impact on the transparency the films. This might be due to the color of curcumin.

#### 2.1.3. Moisture Content and Water Solubility

The moisture content indicates the water holding capacity of the film matrix during drying. The water solubility refers to the disintegration or resistance of the film structure when contacted with water [31]. As presented in Figure 1A, the moisture content did not have significant changes with curcumin addition (*p >* 0.05), which was different from the results of Gao et al. [20]. Generally speaking, with the addition of curcumin, more hydrophilic polar sites (such as phenol hydroxyls) were available to absorb and retain moisture. However, in our study, the porous structure of PCCF might give the moisture an evaporating surface and tunnels to escape.

As shown in Figure 1B, 3 mg of curcumin addition (PCCF-1) had little effect on the solubility of the films (51.9%) compared with the control (53.65%), but 6 mg (PCCF-2) or 9 mg (PCCF-3) of curcumin addition could significantly decrease the solubility (which reached 45.27% and 45.01%, respectively) (*p* < 0.05). This might be due to the exposure to the hydrophobic benzene ring of curcumin, which could increase the solubility resistance.

#### 2.1.4. Mechanical Properties

In general, the properties of films depend on their structure, which relies on their composition and formation process. The addition of curcumin increased the tensile strength (Figure 1C) but decreased the elongation at break (Figure 1D). The tensile strength of PCCF-0 was 3.11 MPa, in good agreement with the result of a previous report [20]. The tensile strength increased significantly to 3.75 MPa when up to 9 mg of curcumin was added (PCCF-3), which was 1.2-fold of that of PCCF-0. It indicated that curcumin might have interaction with chitosan and/or pectin and obtain a more complex structure. The elongation decreased from 15.04% to 10.01% when the addition of curcumin increased to 9 mg, indicating that the PCCF-3 was brittle and not flexible compared with pectin/chitosan film (PCCF-0).

### 2.2. Structure Characterization of PCCF

#### 2.2.1. Differential Scanning Calorimetry (DSC)

The DSC thermogram of PCCF is shown in Figure 2. All films exhibited three distinctive weight loss steps. The thermograms of all films exhibited an initial endothermic peak from 20 to 400 °C. At around 60 °C, the thermogram showed a smaller endothermic peak, which was associated with water release from the sample. Meanwhile, the heat absorption peak moved towards higher temperatures, indicating that the increase of hydrophilic groups could improve the water-polymer interaction and water holding capacity of the films [32]. The melt of crystalline state was manifested by a broad endothermic peak near 146 °C [33]. Additionally, the peak at 288 °C was attributed to the degradation of the PCCF backbone [34]. Besides, the endothermic peak shifted to a higher temperature with the increase of curcumin content, indicating that the curcumin could increase the complexity of the film structure.

#### 2.2.2. X-ray Diffractometry (XRD)

XRD analysis was used to explore the crystalline structure and evaluate the interaction of major film components. The XRD patterns of all films were shown in Figure 3. Each crystalline substance shows its unique diffraction patterns when the X-ray passed through the crystal. The diffraction pattern will show a simple superposition of characteristic diffraction peaks if there is not any interaction or weak interaction among the film matrix. It was observed that PCCF-0 displayed a typical peak of chitosan at 24°. However, the main diffraction angle (2Ɵ) at 10.5° and 18° of chitosan as well as the broad diffraction peak at 10° which was assigned to the characteristic of pectin were not present [35]. It was supposed that the carboxyl group of pectin bonded with the amino group of chitosan, and the network structure disturbed the original crystallization arrangement of the chitosan chain. With the addition of curcumin, the XRD pattern showed a stable baseline and sharp peaks at 8.97°, 14.63°, 17.6° and 23.59°, respectively, which were the characteristic diffraction peaks of curcumin [36]. The other diffraction peaks were weak or disappeared gradually, indicating that the curcumin had been well dispersed and crosslinked to the biopolymer matrix.

#### 2.2.3. Fourier Transform Infrared Spectroscopy (FT-IR)

The Fourier transform infrared spectroscopy (FT-IR) spectrum is shown in Figure 4 and the analyzed data are presented in Table 2. The PCCF exhibited typical characteristic bands around 3380 and 3410 (overlapping of O–H and N–H stretching vibrations, amide-A), 2935 (C–H stretching, amide-B), 1733 (amide Ι, C=O stretching vibration) and 1595 cm^−1^ (amide Π, –NH bending vibration) [37]. The absorption around 3448 (stretching vibration of O–H and N–H) and 1595 cm^−1^ (stretching of amide Π) shifted to a lower wavenumber and the new absorption (3411 cm^−1^) arose, it might due to the electrostatic interaction and hydrogen bonds between curcumin and pectin-chitosan films [37]. It has been reported that the amide A and amide II bands might shift and show displacement by interactions between hydroxyl groups of pectin or amino groups of chitosan and polyphenol (curcumin) [38]. In addition, relevant studies have shown that the strong hydrogen bond between phenolic compounds and polysaccharides could reduce the original chemical bond constant of polysaccharides, causing the absorption frequency to move towards a lower wavenumber [39].

#### 2.2.4. Scanning Electron Microscope (SEM)

The morphological characteristics of the surface of PCCF were observed by SEM (Figure 5). The surface of PCCF-0 was compact, indicating the excellent film formability of pectin and chitosan. The surface of PCCF also had some white spots, which had been reported that the calcium ions could accumulate and form white patches in the polysaccharide film [40]. Meanwhile, some convex pieces and particles were observed when the curcumin was added to the film, it might be because the hydroxyl group in chitosan and curcumin could dehydrate and condense with the carboxyl group in pectin. The complex reaction resulted in the formation of a complex three-dimensional network structure. With the curcumin addition up to 9 mg (PCCF-3), some flower-like crystals appeared on the surface of the film, which have been identified as curcumin crystals, thus it could act as a barrier for water diffusion (Figure 5D) [41].

### 2.3. Antioxidant Activity

Antioxidant activity is very important for food packaging films since free radicals could lead to discoloration, rancidity and odor in packaged foods [42]. As shown in Figure 6A,B), PCCF exhibited free radical scavenging activity over time and tended to be stable after 8 h. The ABTS free radical scavenging activity of PCCF-3 at 24 h was 29.07 ± 3.26%, 1.35-fold of that of the control (Figure 6A). Meanwhile, PCCF was also observed to possess a strong ability to scavenge DPPH radical at 24 h from 53.29 ± 3.86% to 58.66 ± 3.43%, which was obviously higher than that of control (44.78 ± 3.6%) (Figure 6B). Moreover, the DPPH free radical scavenging activity of PCCF was in a dose-dependent manner. It has been reported that curcumin has large amounts of phenolic hydroxyls, which can effectively provide hydrogen donors to free radicals, thus blocking the chain reaction, and improve the antioxidant activity [41]. Results showed that PCCF could inhibit free radical chain reactions and could reduce the oxidative deterioration of food, especially in oil-rich foods (such as meat, milk and cakes).

### 2.4. Antiseptic Capacity in Food System

Microbial contamination is the main pathway of food spoilage and the main cause of food safety problems. Although some chemical preservatives (potassium sorbate and sodium benzoate) have been used to prevent the growth of food-borne bacteria, their potential toxicity is attracting more and more attention. PCCF was expected to have a potential antisepsis capability in the food system, based on its in vitro antioxidant activity. As shown in Figure 7, PCCF showed antiseptic activity in a dose-dependent manner. After 5 days of storage at 25 °C and 45% humidity, the colony-forming unit (CFU) of the control group increased exponentially (the data of log (CFU/g) up to 7.70 ± 0.08). But the CFU of pork covered with PCCF increased, with the data of log (CFU/g) ranged from 6.87 ± 0.09 to 7.14 ± 0.05. The antiseptic properties of the film might be due to the effects of curcumin and chitosan. It has been reported that curcumin had excellent bactericidal properties [24], which might be due to the multi-point interaction between curcumin and bacteria proteins. Curcumin could specifically coagulate structural proteins, bind to deoxyribonucleic acid molecules, destroy the cell membrane and cell wall of microorganisms, and thus inhibit their growth [43]. Furthermore, chitosan could interact with negatively charged residues on the surface of microbial cells through electrostatic attraction, flocculate and adsorb on the surface of microbial, thus blocking the physiological metabolism of microbial and ultimately inhibiting the growth of microbial [44].

## 3. Materials and Methods

### 3.1. Materials and Reagents

Chitosan (deacetylation ≥ 90%) and pectin (purity ≥ 65.0%) were purchased from Yuanye (Shanghai, China). 2,2′-azinobis-3-ethylbenzthiazoline-6-sulphonate (ABTS) and 1,1-diphenyl-2-picrylhydrazyl (DPPH) were obtained from Macklin (Shanghai, China). Gallic acid (purity ≥ 98%) was purchased from Sigma-Aldrich (St. Louis, MO, USA). Curcumin was obtained from Aladdin (Shanghai, China). Fresh pork was purchased from a local supermarket in Guangzhou, China. All other reagents used were analytic grade.

### 3.2. Preparation of PCCF

A procedure for preparing polysaccharide film was depicted in Figure 8. Briefly, the choline chloride, lactic acid and glycerol (14:9:9, *w*/*w*/*w*) were mixed and stirred at 60 °C for 1 h to obtain the transparent liquid (DES, pH = 1.1 ± 0.02) [45]. The pectin (2 g) was dissolved in 0.5% calcium chloride aqueous solution (40 mL) and then stirred for 10 min. The chitosan (1 g) was suspended in deionized water (30 mL) containing DES (1 mL). At this stage, DES provided an acidic medium, which could help to dissolve chitosan. Different amounts of curcumin (0 mg, 3 mg, 6 mg, 9 mg) were added into 30 mL of water (containing 1 mL of DES) followed by ultrasound (KW-800KDE, Shufeng Co., Kunshan, China) at 100 W for 10 min. Then, chitosan solution and curcumin solution were successively added into the prepared pectin solution to obtain the pectin/chitosan film with different concentrations of curcumin (named as PCCF-0 (0 mg), PCCF-1 (3 mg), PCCF-2 (6 mg), PCCF-3 (9 mg), respectively). The mixture was stirred for 2 h at room temperature and then poured into a plastic petri dish with a radius of 6 cm (20 mL mixture per dish). Subsequently, all dishes were dried at 60 °C for 6 h (PH-050(A), Qixin Co., Shanghai, China). The peeled films were kept in a drying vessel for further use.

### 3.3. Characterization of PCCF

#### 3.3.1. Thickness

The thickness of the films was determined by using a digital micrometer (METR IS0-GEW, Meinaite Co., Shanghai, China). The thickness was calculated as the average of 10 random positions of each sample.

#### 3.3.2. Film Color and Optical Characterization

The film color parameters were determined using a portable colorimeter (WSC-S, Yidian Co., Shanghai, China), at five random locations of the films. The L* (0 = black, 100 = white), a* (negative value means greenness, while positive value is redness) and b* (negative value means blueness, while positive is yellowness) values of films were recorded, with the white reference plate as the background.

Light transmission was analyzed according to the method of Ezati et al. [30,46]. The films were cropped into stripes (1 cm × 3.5 cm) and vertically put in the quartz cuvette of the spectrophotometer (752 N, Shanghai Precision and Scientific Instrument Co., Ltd., Shanghai, China). An empty cuvette was used as the blank control. The transmittance of the film was measured at 660 nm.

#### 3.3.3. Determination of Moisture Content and Water Solubility

The films were weighed as m_1_, then they were dried at 105 °C until a constant weight (m_2_). The moisture content was determined by the following equation [35]: Moisture content (%) = (m_1_ − m_2_)/m_1_ × 100% (1).

In the water solubility assay [28], the weight of the films was recorded as W_1_. The samples were dried at 105 °C to constant weight and then placed in a glass petri dish with 20 mL distilled water and kept for 24 h. When the settled time elapsed, the solution in the plate was sucked slowly with a pipette. The film residues were dried at 105 °C to constant weight then weighted (W_2_). The water solubility (%) = (W_1_ − W_2_)/W_1_ × 100% (2).

#### 3.3.4. Mechanical Properties

TA-XT plus-C texture analyzer (Stable Micro Systems, TA-XT PlusC, Surrey, UK) was used to evaluate the mechanical properties, including tensile strength and elongation at break. The film was prepared as the same size slices (90 mm × 15 mm). The distance between the initial grips was set as 60 mm, and the stretching speed of the upper grip was set as 5 mm/s, respectively. Tensile strength (MPa) = (F (N))/(thickness × width (mm))(3), Elongation at break (%) = ΔL/L_0_ (4) [35], where F was the stress for film fracture (N), ΔL and L_0_ were the elongated and initial lengths (mm) of film, respectively.

### 3.4. Differential Scanning Calorimetry (DSC)

The thermal properties of films were analyzed by a differential scanning calorimeter (DSC 250, TA instruments, New Castle, DE, USA). The sealed aluminum pans containing samples (approximately 10 mg) were heated from 20 °C to 400 °C at a rate of 20 °C/min with a 50 mL/min nitrogen flow [32]. An empty sealed aluminum pan was taken as the reference.

### 3.5. X-ray Diffractometry (XRD)

XRD pattern of films was measured by D8 Advance X-ray diffractometer (Bruker AXS GmbH, Karlsruhe, Germany) at 2Ɵ diffraction angle from 20° to 60°. The surface of the film should be as smooth as possible.

### 3.6. Fourier Transform Infrared Spectroscopy (FT-IR)

The Fourier transform infrared spectroscopy (FT-IR) spectra of the films were recorded using an FT-IR spectrophotometer (Tensor 27, Bruker Co. Ltd., Bergisch Gladbach, Germany). Discs of samples with KBr were scanned in the range of 400–4000 cm^−1^ with a resolution of 4 cm^−1^.

### 3.7. Scanning Electron Microscope (SEM)

The microstructures of the surface of films were characterized by using a scanning electron microscope (ZEISS EVO 18, Cambridge, UK). The sample was coated with a gold-palladium alloy and photographed at a magnification of 200× and 5000×.

### 3.8. Antioxidant Activity

PCCF (1 g) was immersed in deionized water (30 mL) and sealed storage at 25 °C. Thereafter, the film soak solution was collected and determined their ABTS and DPPH free radicals scavenging activities, respectively.

ABTS (7 mM, 7.5 mL) solution and potassium persulphate (140 mM, 132 μL) were mixed and stored in the dark for 16 has a stock solution. Then the ABTS solution was diluted with anhydrous ethanol to obtain an absorbance of 0.7 ± 0.02 at 734 nm (SpectraMax190 Microplate Reade, Molecular Devices, Downingtown, PA, USA). The film soak solution (50 μL) and diluted ABTS solution (100 μL) were mixed and stored for 10 min in the dark. The same ABTS solution without film soak solution was used as the control. After incubation, the absorbance was recorded at 734 nm. ABTS free radical scavenging activity (%) = (1 − A_sample 734_/A_control 734_) × 100% (5) [21].

DPPH solution (1 mL, 0.1 mM dissolved with anhydrous ethanol) was mixed with the film soak solution (1 mL) and kept in the dark for 30 min. The absorbance of the mixture was measured at 517 nm. The DPPH solution without film soak solution was used as the control. DPPH free radical scavenging activity (%) = (1 − A_sample 517_/A_control 517_) × 100% (6) [21].

### 3.9. Antiseptic Capacity in Food System

The antiseptic capacity of PCCF was determined in a fresh pork system. Pork pieces (25 mm × 25 mm) with a weight of 2.5 g were placed in sterile Petri dishes and covered with PCCF (6 cm in diameter). The pork without film cover was prepared as the control. The dishes were then sealed with polyethylene film and incubated at 25 °C and 45% relative humidity environment. Meat samples were homogenized and diluted with sterile water to an appropriate concentration to obtain a meat homogeneous solution. Then 1 mL of meat homogeneous solution was mixed with PCA medium, in order to determine viable cell counts of microorganism. The number of colonies was counted after incubation at 37 °C for 24 h [18].

### 3.10. Statistic Analysis

Data were expressed as mean ± SD from at least triplicate determinations. The analysis of Variance (ANOVA) was performed by SPSS (Version 25.0, Windows, SPSS Inc., Chicago, IL, USA), and the graph base test was used to determine significant differences among mean values (*p* < 0.05).

## 4. Conclusions

In conclusion, a novel polysaccharide-based film (PCCF) was successfully prepared by using pectin, chitosan and curcumin. The curcumin was observed to cross-link with the pectin/chitosan film. The addition of curcumin could decrease the water solubility, elongation and opacity of PCCF significantly, but increase the tensile strength. The PCCF also showed sustained antioxidant and antiseptic capacity. Our studies suggest that curcumin addition might possess potential value to be used as a packaging film in food industries.

## Figures and Tables

**Figure 1 molecules-26-02152-f001:**
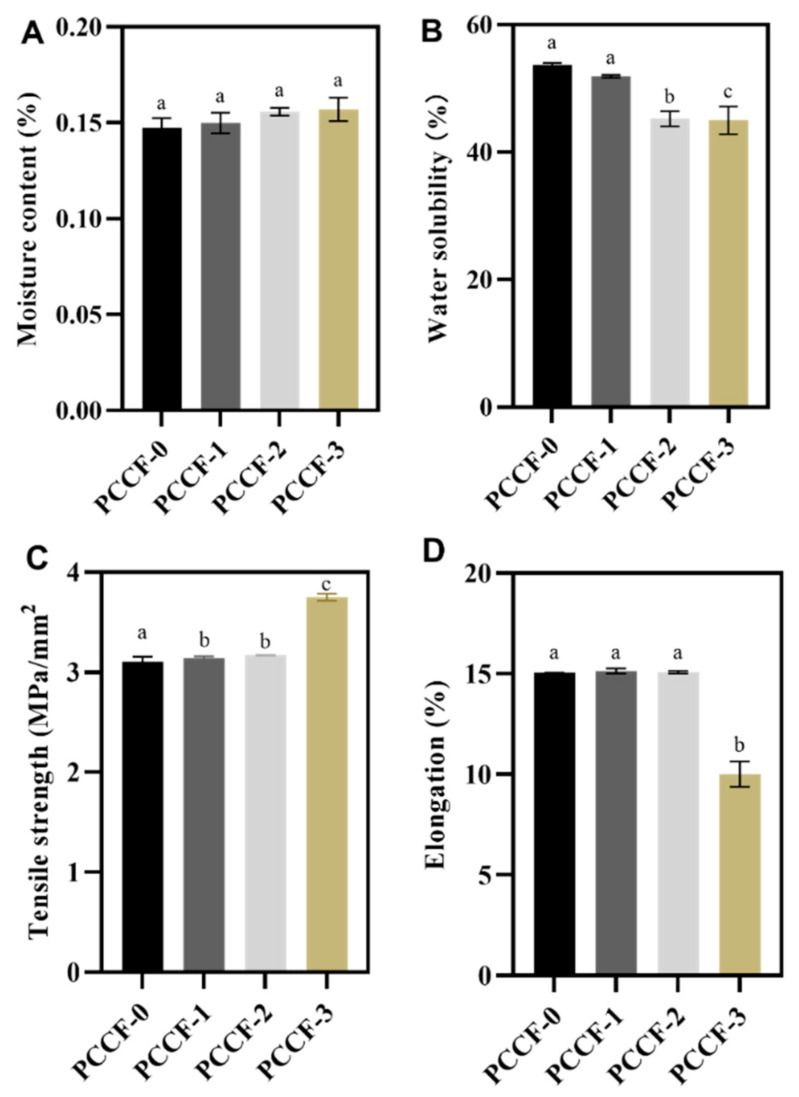
The physical properties of PCCF: (**A**) moisture content; (**B**) water solubility; (**C**) tensile strength; (**D**) the elongation of PCCF. Different letters denote statistically significant differences.

**Figure 2 molecules-26-02152-f002:**
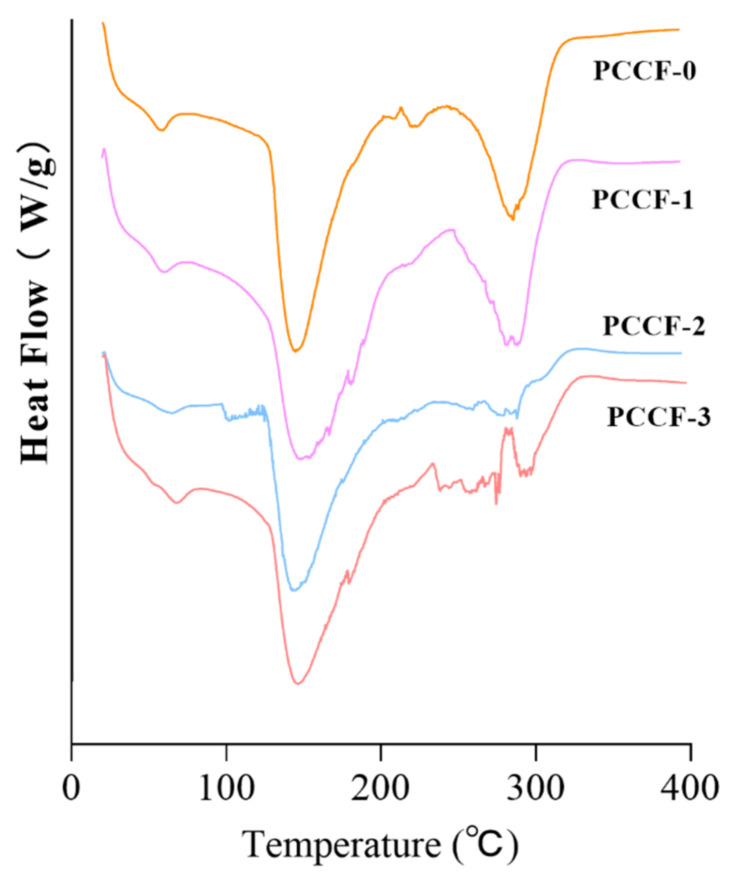
The DSC thermograms of PCCF.

**Figure 3 molecules-26-02152-f003:**
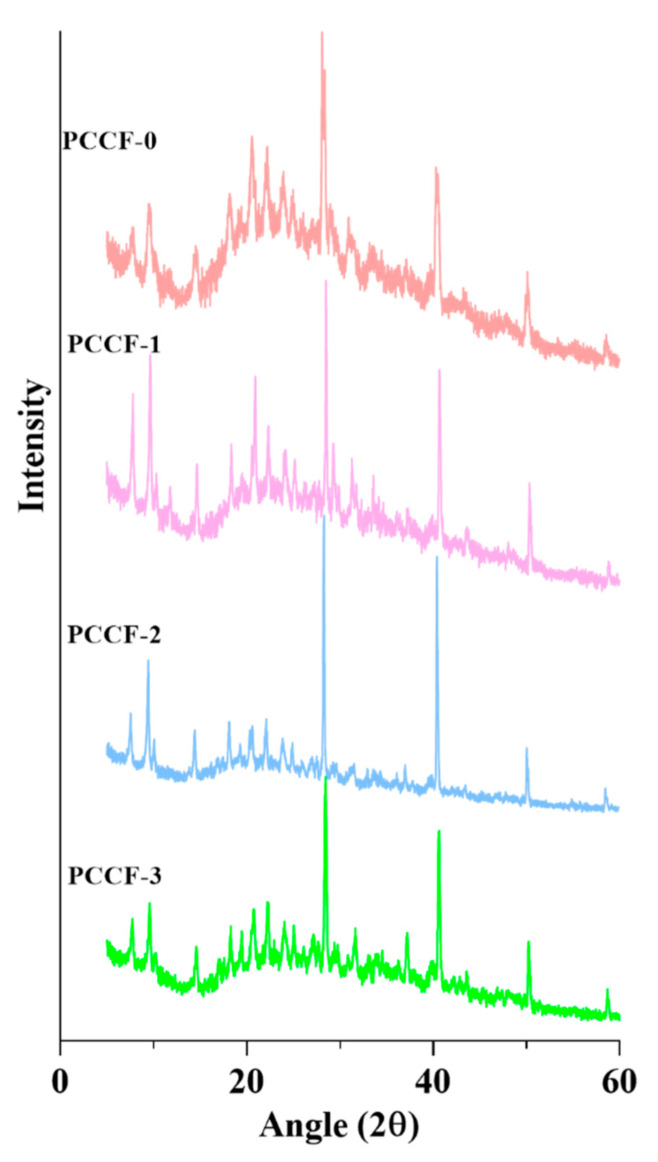
X-ray diffractograms of PCCF.

**Figure 4 molecules-26-02152-f004:**
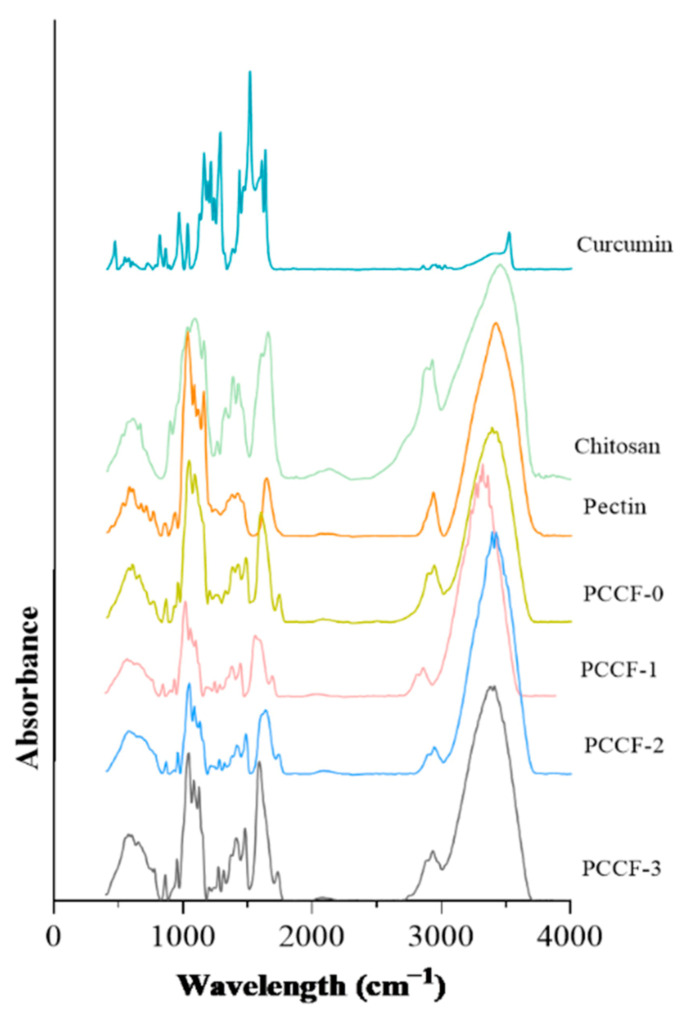
The Fourier transform infrared (FT-IR) spectrum of PCCF.

**Figure 5 molecules-26-02152-f005:**
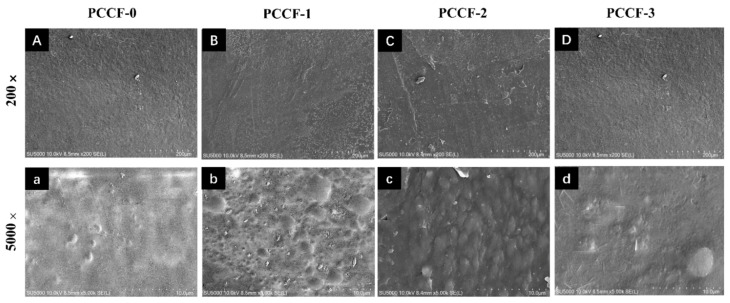
Microstructure and surface morphology of PCCF observed by SEM at a magnification of 200× (**A**–**D**) and 5000× (**a**–**d**).

**Figure 6 molecules-26-02152-f006:**
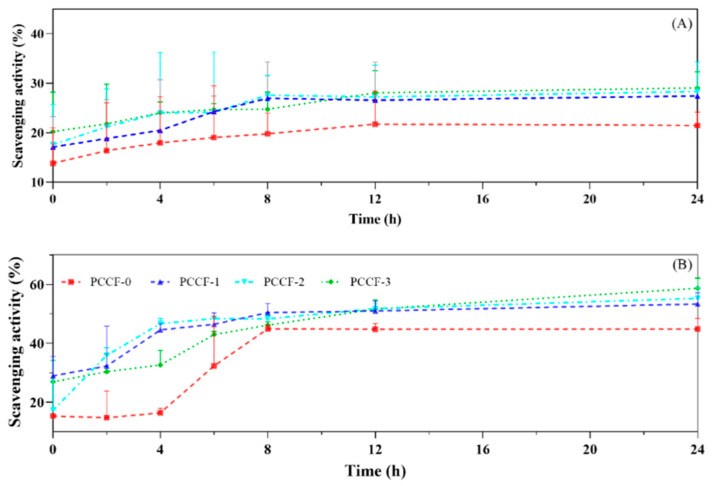
Antioxidant activity of PCCF (**A**) ABTS free radical scavenging activity; (**B**) DPPH free radical scavenging activity.

**Figure 7 molecules-26-02152-f007:**
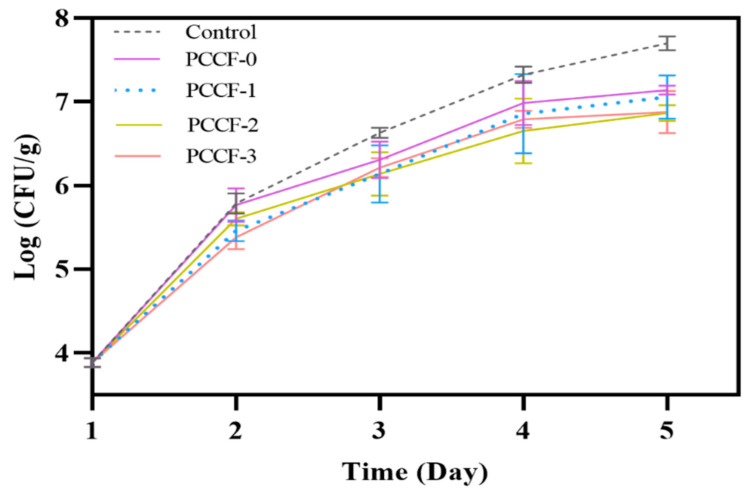
The antiseptic activity of PCCF in food system.

**Figure 8 molecules-26-02152-f008:**
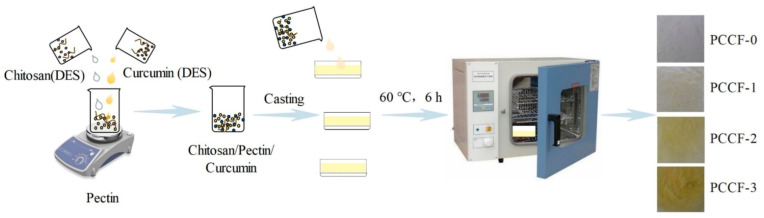
Procedure for preparation of pectin-chitosan-curcumin film (PCCF).

**Table 1 molecules-26-02152-t001:** The physical and optical characterization of PCCF.

Sample	Thickness (mm)	Transparency (%)	L*	a*	b*
PCCF-0	0.186 ± 0.019 ^d^	30.27 ± 2.87 ^a^	49.64 ± 0.96 ^d^	4.31 ± 0.41 ^a^	30.365 ± 0.47 ^d^
PCCF-1	0.328 ± 0.035 ^c^	18.08 ± 3.02 ^b^	52.78 ± 0.66 ^c^	2.26 ± 0.29 ^b^	38.17 ± 1.3 ^c^
PCCF-2	0.420 ± 0.035 ^b^	9.22 ± 0.01 ^c^	54.98 ± 1.16 ^b^	1.90 ± 0.31 ^b^	38.39 ± 0.63 ^b^
PCCF-3	0.426 ± 0.025 ^a^	7.72 ± 0.81 ^d^	57.82 ± 0.07 ^a^	0.56 ± 0.07 ^b^	44.86 ± 0.44 ^a^

Each value is expressed as mean ± SD (*n* = 3). Different superscript letters denote statistically significant differences (*p* < 0.05).

**Table 2 molecules-26-02152-t002:** Wavenumber and IR assignment of PCCF.

Sample	IR Assignment				
	NR2, –NH, O–H Stretching	C–H Stretching	C=O Stretching	C=C Stretching, N–H Bending Stretching	C=C Stretching, C–OH Stretching
PCCF0	3381	2887	1735	1595	1479
PCCF1	3448, 3419, 3409, 3388	2935	1733	1595	1479
PCCF2	3411, 3382	2935	1733	1595	1481
PCCF3	3413, 3379	2935	1733	1593	1481

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
