# Peer review of "Effect of Curcumin Addition on the Properties of Biodegradable Pectin/Chitosan Films"

_molecules, 2021, doi:10.3390/molecules26082152_

Round 1
Reviewer 1 Report
The work of Xie et al. presents a novel approach to produce a pectin/chitosan film, enhancinig the bioactivity thanks to the addition of curcumin. The process applied green approaches such as DES and US. The different films have been characterized by means of several techniques, investigating physical properties and bioactivity.
The work appears generally consistent, facing an actual and interesting topic. Anyhow, it is opinion of this referee that before to be eligible of pubblication on Molecules, some inaccuracies should be improved. Furthermore, some references should be implemented in the work to support a few points.
Hereafter some considerations:
The authors do not face the effect of DES on the film formulation. Some considerations should be provided across the paper, both in the discussion and in the conclusion. In the introduction it is reported that DES enhanced the mechanical properties of the product, but then not other details/considerations have been made concerning this point.
Line 4: please, remove “and” at the beginning of the sentence.
Line 31: The FAO reference shouldn’t be inserted as a direct link in the text, but reported in the bibliography (numbered as [1]).
Line 34: Please, change “prolong” with “extend”.
Line 54: Vapour typo.
Line 69: The reference 12 does not appear appropriate for the subject. Authors listed a large asset of DES applications, so it is required at least one updated reference for every topic (bioactive extraction, reaction medium, melt processing, organic synthesis). Hereafter some suggestions:
Bioactive extraction: Grillo, G.; Gunjević, V.; Radošević, K.; Radojčić Redovniković, I.; Cravotto, G. Deep Eutectic Solvents and Nonconventional Technologies for Blueberry-Peel Extraction: Kinetics, Anthocyanin Stability, and Antiproliferative Activity. Antioxidants 2020, 9, 1069, doi:10.3390/antiox9111069.
Reaction medium and organic synthesis: Piemontese, L.; Sergio, R.; Rinaldo, F.; Brunetti, L.; Perna, F.M.; Santos, M.A.; Capriati, V. Deep Eutectic Solvents as Effective Reaction Media for the Synthesis of 2-hydroxyphenylbenzimidazole-Based Scaffolds en Route to Donepezil-Like Compounds. Molecules 2020, 25, 574. https://doi.org/10.3390/molecules25030574.
Melt processing: Grylewicz, A.; Spychaj, T.; Zdanowicz, M.; Thermoplastic starch/wood biocomposites processed with deep eutectic solvents, Composites Part A: Applied Science and Manufacturing, 2019, 121, 517-524, https://doi.org/10.1016/j.compositesa.2019.04.001.
Line 73: typo, “heteropolysaccharide”.
Line 86: According to the paper topic, among the curcumin properties, I suggest to add also antibacterial, antifungal and antiviral activities. Hereafter, some useful references to support and reinforce the activity statements:
Sueth-Santiago, V.; Mendes-Silva, G.P.; Decoté-Ricardo, D.; de Lima, M.E.F. Curcumin, the golden powder from turmeric: Insights into chemical and biological activities. Quim. Nova. 2015, 38, 538–552. doi:10.5935/0100-4042.20150035.
Bessone, F.; Argenziano, M.; Grillo, G.; Ferrara, B.; Pizzimenti, S.; Barrera, G.; Cravotto, G.; Guiot, C.; Stura, I.; Cavalli, R.; et al. Low-dose curcuminoid-loaded in dextran nanobubbles can prevent metastatic spreading in prostate cancer cells. Nanotechnology 2019, 30, 214004–214016, doi: 10.1088/1361-6528/aaff96
Moghadamtousi, S.Z.; Kadir, H.A.; Hassandarvish, P.; Tajik, H.; Abubakar, S.; Zandi, K.; A Review on Antibacterial, Antiviral, and Antifungal Activity of Curcumin, BioMed Res. Int., 2014, 2014, 1-12. https://doi.org/10.1155/2014/186864
Line 87: It is not clear to this referee what the authors means with “intestinal products”.
Line 91: “little work” appear to be too conversational. Please change with “not many efforts have been done…”.
Line 96: “Characterized in “characterize”
Line 113: Please, uniform the digit number.
Line 118-125: In this section the word “film” appear too many times. For sake of readability, this referee suggest to rephrase accordingly.
Line 126: The table should be self-standing, hence the caption should report information of the statistical values (i.e. superscripts).
Line 129: Please, remove “and” at the beginning of the sentence.
Line 142: “dependent” in “depend”.
Line 164: Are there any references supporting the assignments of the DSC peaks?
Line 178: Please modify “disappeared” with “not present”.
Line 179: The specification of “hydrogen” is redundant.
Figure 4: A translation is missing.
Table 2: It is my opinion that the table could be more clear if the “IR assignments” were positioned as first row.
Line 209-210: This sentence should be rephrased for sake of clarity.
Line 212-213: It is not clear what the authors mean with “the hydroxyl group in chitosan and curcumin could dehydrate and condense with the carboxyl group in pectin”. If you refer to the esterification reaction, it is advisable to better explain the point.
Figure 5: The dimension of the imagines is not enough to allow a proper observation. Please, provide larger ones. Furthermore, it is fundamental for the marker to be clear and visible for microscopies.
Line 228: Please, reformulate “concentrate-dependent” in “dose-dependent”.
Line 269: transparent instead of transparency.
Line 272: Authors should provide some pH measurements to support the statement. The addition of 1 mL of DES made from choline chloride, lactic acid and glycerol could be not enough acid to modify consistently the solution acidity.
Line 274: Please, for sake of clarity, provide the US type (horn, bath, cup-horn…).
Figure 8: Poor quality. Please, provide a better one.
Line 286: Please, provide band and model of the micrometer.
Line 304: An hypertext link is present.
Line 307, 313, 345, 350: Please, report Water Solubility, Tensile strength, Elongation at break, ABTS free radical scavenging activity and DPPH free radical scavenging activity as Equation (1, 2, ..), according to Journal template instruction.
Reviewer 2 Report
Article: Effect of curcumin addition on the properties of Pectin/chitosan biodegradable film
molecules-1157847
Review comments:
- What is the scientific novelty of the manuscript? Please, a separate section devoted to the innovative potential of the conclusions obtained should be explained and added to the manuscript.
- The conclusion section needs to be strengthened.
- Polish the manuscript and carefully correct grammatical mistakes- should be thoroughly checked for language errors.
Reviewer 3 Report
This paper reports the addition of curcumin on the properties of Pectin/chitosan biodegradable film used for food package. It is an interesting research topic and the information could be useful in the related food industry.
Comments:
- In Table 1, what is the unit for the L, a and b?
- In Fig. 1B, what is the unit for water solubility?
- Experiments on the biodegradability of the PCCF films are required.
- It is necessary to add positive controls using commercial food package film in the experiments on mechanical properties and antiseptic activity of PCCF.
- The English in whole manuscript is not sound, please improve it by a professional English write.
Round 2
Reviewer 3 Report
The authors addressed all my comments in the revised MS.